Bacterial curli protein promotes the conversion of PAP248-286 into the amyloid SEVI: cross-seeding of dissimilar amyloid sequences

Hartman Kevin 1 2
Brender Jeffrey R. 1 2
Monde Kazuaki 3 5
Ono Akira 3
Evans Margery L. 4
Popovych Nataliya 1 2 6
Chapman Matthew R. 4
Ramamoorthy Ayyalusamy ramamoor@umich.edu 1 2
1 Department of Chemistry , University of Michigan , USA
2 Department of Biophysics , University of Michigan , USA
3 Department of Microbiology and Immunology , University of Michigan Medical School , USA
4 Department of Molecular, Cellular, and Developmental Biology , University of Michigan , USA
Pastore Annalisa
5 Present address: Department of Medical Virology, Faculty of Life Sciences, Kumamoto University, Japan

6 Present address: Genentech, USA

Electronic publication date: 2013 Feb 12
Publication date: 2013
Volume: 1
Electronic Location ID: e5
Received 2012 Nov 14; Accepted 2012 Dec 5
Copyright: © 2013 Hartman et al.
Copyright year: 2013
Copyright holder: Hartman et al.
License: This is an open access article distributed under the terms of the Creative Commons Attribution License, which permits unrestricted use, distribution, and reproduction in any medium, provided the original author and source are credited.
License URL: https://creativecommons.org/licenses/by/3.0/

Keywords: Functional amyloid, Strain, Kinetics, Seeding, HIV

Funding: NIH GM095640 AI089282 AI073847 This study was supported by research funds from the NIH (GM095640 to Ayyalusamy Ramamoorthy; AI089282 to Akira Ono; and AI073847 to Matthew R. Chapman). The funders had no role in study design, data collection and analysis, decision to publish, or preparation of the manuscript.

==============================
Fragments of prostatic acid phosphatase (PAP248-286) in human semen dramatically increase HIV infection efficiency by increasing virus adhesion to target cells. PAP248-286 only enhances HIV infection in the form of amyloid aggregates termed SEVI (Semen Enhancer of Viral Infection), however monomeric PAP248-286 aggregates very slowly in isolation. It has therefore been suggested that SEVI fiber formation in vivo may be promoted by exogenous factors. We show here that a bacterially-produced extracellular amyloid (curli or Csg) acts as a catalytic agent for SEVI formation from PAP248-286 at low concentrations in vitro, producing fibers that retain the ability to enhance HIV (Human Immunodeficiency Virus) infection. Kinetic analysis of the cross-seeding effect shows an unusual pattern. Cross-seeding PAP248-286 with curli only moderately affects the nucleation rate while significantly enhancing the growth of fibers from existing nuclei. This pattern is in contrast to most previous observations of cross-seeding, which show cross-seeding partially bypasses the nucleation step but has little effect on fiber elongation. Seeding other amyloidogenic proteins (IAPP (islet amyloid polypeptide) and Aβ1−40) with curli showed varied results. Curli cross-seeding decreased the lag-time of IAPP amyloid formation but strongly inhibited IAPP elongation. Curli cross-seeding exerted a complicated concentration dependent effect on Aβ1−40 fibrillogenesis kinetics. Combined, these results suggest that the interaction of amyloidogenic proteins with preformed fibers of a different type can take a variety of forms and is not limited to epitaxial nucleation between proteins of similar sequence. The ability of curli fibers to interact with proteins of dissimilar sequences suggests cross-seeding may be a more general phenomenon than previously supposed.

Introduction

Over 33 million people worldwide are currently infected with the HIV virus. While HIV is readily transmitted in vivo, HIV is a surprisingly ineffective pathogen in vitro (Thomas, Ott & Gorelick, 2007). The key barrier to effective viral transmission is attachment to the target cell surface. The short-half life of almost all viruses in solution (∼6 h for HIV) ensures that infection can only be achieved only if a significant number of viral particles adhere to the cell surface in a relatively narrow time window upon exposure (Perelson et al., 1996). A peptide fragment naturally occurring in human seminal fluid (PAP248-286) facilitates this first step, and in doing so, greatly enhances the infectivity of HIV (Munch et al., 2007). When fibrillized into amyloid fibers termed SEVI (Semen-derived Enhancer of Virus Infection), (Munch et al., 2007; Arnold et al., 2012; Roan et al., 2011) PAP248-286 bridges the membranes of the HIV virion and target cells (Munch et al., 2007; Roan et al., 2009; Roan et al., 2010; Wurm et al., 2010; Kim et al., 2010). The result is a dramatic increase in the infectivity of the HIV virus. While up to 100,000 HIV virions are required to establish infection in the absence of SEVI amyloid fibers, only 3 are needed in its presence (Munch et al., 2007).

This enhancement is specifically linked to the SEVI amyloid conformation of PAP248-286 as monomers of PAP248-286 have little effect (Munch et al., 2007; Roan et al., 2010; Capule et al., 2012; Olsen et al., 2010). Since PAP248-286 only enhances HIV infection when in the aggregated SEVI amyloid form, molecules that initiate fibrillization of PAP248-286 can indirectly increase HIV infectivity. While amyloid formation is typically energetically favorable, the actual rate of formation can be very slow. Specifically, amyloid formation is a nucleation-dependent process in which a slow rate-limiting nucleation step is followed by the faster process of extension of the fiber from the nuclei. PAP248-286 is subject to significant proteolytic degradation in seminal plasma unless it is in the SEVI amyloid form (Martellini et al., 2011). The total production of SEVI is therefore ultimately controlled by the rate of amyloidosis due to the proteolytic degradation of unfibrillized PAP248-286. PAP248-286 is very slow to fibrillize in the absence of shaking in vitro, (Ye et al., 2009; Olsen et al., 2012) which suggests SEVI production will be very limited in vivo. Nevertheless, the SEVI amyloid form is found in semen at concentrations of up to 35 µg/mL (Munch et al., 2007; Kim et al., 2010). This finding suggests additional cofactors may be present that accelerate SEVI formation from PAP248-286 before the PAP248-286 monomer is degraded by the cell’s proteolytic machinery (Olsen et al., 2012).

One set of likely cofactors are other cellular proteins, since the factors that drive self-association of amyloidogenic proteins also tend to favor promiscuous binding to a variety of other proteins (Olzscha et al., 2011; Manzoni et al., 2011). In particular, amyloidogenic proteins frequently have the ability to cross-polymerize, that is amyloid fibers from one protein can catalyze the formation of amyloid fibers from another amyloidogenic protein. Cross-seeding amyloid fibrillogenesis in this manner can dramatically enhance the kinetics of amyloid formation by providing preformed nuclei for further aggregation. Furthermore, the final amyloid fiber can in some cases takes on some of the characteristics of fibers from the initial seeding amyloid protein (Dzwolak et al., 2005; Foo et al., 2011; Cloe et al., 2011).

Amyloid nucleation is often considered a highly specific process dependent on a high degree of similarity in both peptide sequence and fiber morphology between the seed and the protein being nucleated. This conclusion has largely been motivated by research on prion amyloid fibers, where the inability of highly homologous prions to cross-seed amyloid formation presents an inter-species barrier to prion transmission (Chien, Weissman & DePace, 2004; Apostol et al., 2011). However, this view has recently been challenged by the observation of efficient cross-polymerization between several non-prion amyloid species (Yagi et al., 2005; Han, Weinreb & Lansbury, 1995; Westermark, Lundmark & Westermark, 2009; Morales et al., 2010; O’Nuallain et al., 2004; Ma & Nussinov, 2012). Seeds formed from amyloid fibers of some non-homologous proteins can either reduce or eliminate the lag-time of amyloid formation of another protein, (Yagi et al., 2005; Han, Weinreb & Lansbury, 1995; Westermark, Lundmark & Westermark, 2009; Morales et al., 2010; O’Nuallain et al., 2004; MacPhee & Dobson, 2000) although the phenomenon is not universal and even single point mutations have been shown to disrupt cross-seeding in some cases (O’Nuallain et al., 2004; Jarrett & Lansbury, 1992; Hinz, Gierasch & Ignatova, 2008; Rajan et al., 2001). Cross-seeding may also be asymmetric, with one protein able to seed another but not be seeded by it (Yu et al., 2012; Siddiqua et al., 2012).

The amyloids produced by many bacteria and fungi are of particular interest in considering possible cross-seeding reactions with PAP248-286 because of the high incidence of microbial infection in the vagina and the frequent coexistence with bacterial and fungal infection with HIV infection (Thurman & Doncel, 2010). A particularly well studied example is a highly amyloidogenic protein secreted by Escherichia coli (E. Coli) and related bacteria, CsgA, that polymerizes into curli fibers that are involved in cell-to-cell and cell-to-surface interactions (Jordal et al., 2009; Barnhart & Chapman, 2006). Although curli fibers have not to our knowledge been directly detected so far in semen or the vagina, curli and curli-like amyloid fibers are ubiquitous in mammalian hosts (Cegelski et al., 2009). In fact, the innate immune response invoked by almost all amyloids has been proposed to have evolved as a response to curli amyloid formation by E. Coli (Tukel et al., 2009). Since functional amyloid fibers from bacteria or yeast similar to curli may be colocalized with PAP248-286 at the initial site of HIV infection, we tested the effect of preformed curli amyloid fibrils on the kinetics of SEVI fiber formation. We found curli does significantly enhance the rate of SEVI fibrillization, although through an unusual mechanism for a cross-seeding interaction.

Methods & materials

Peptide preparation

PAP248-286 was obtained from Biomatik Corporation. To prepare monomeric PAP248-286, lyophilized PAP248-286 was first quickly dissolved in 20% acetic acid to a final concentration of 1 mg/ml to remove preformed aggregates. The aggregate free solution was then lyophilized overnight. A 3 mg/ml stock solution was made from the lyophilized peptide in the assay buffer (10 mM phosphate buffer, pH 7.3 containing 150 mM NaCl).

Human IAPP was obtained from Genscript. Monomeric IAPP was prepared by first dissolving in hexafluoroisopropanol to a concentration of 1 mg/ml then lyophilizing overnight. Aβ1−40 was obtained from Anaspec. Monomeric Aβ1−40 was prepared in a similar way using 2% ammonium hydroxide instead of hexafluoroisopropanol. The lyophilized pellet of both peptides was dissolved directly in the assay buffer.

CsgA and CsgB preparation

CsgA and CsgB were expressed (Hammer, Schmidt & Chapman, 2007) and purified (Wang et al., 2007) as previously described. Briefly, both were overexpressed as His-tag fusion proteins in LSR12 bacteria. Following centrifugation for 20 min at 5000 × g, the bacterial pellet was incubated in 8 M guanidine hydrochloride (from Sigma, adjusted to pH 7.2 by the addition of NaOH) for 24 h with stirring. After incubation, the cells were centrifuged again for 20 min at 10,000 × g and the supernatant sonicated. The solution was then passed over a HIS-Select™ HF nickel-nitrilotriacetic acid column, washing first with 50 mM KPi (pH 7.3) followed by 12.5 mM imidazole in 50 mM KPi (pH 7.3) to remove non-specifically bound proteins and then with 125 mM imidazole in 50 mM KPi (pH 7.3) to elute CsgA or CsgB.

To prepare amyloid fibers of CsgA and B, the curli containing fractions from the HIS-Select™ column allowed to fibrillize overnight at room temperature. Protein concentration was measured by the BCA (Bicinchoninic Acid) assay prior to fibrillization. The resulting fibers were then centrifuged (15 min at 10,000 × g) and washed with water to remove the imidazole salt. This procedure was repeated four times and the pellet then lyophilized. The lyophilized pellets were then reconstituted in the assay buffer and sonicated with a probe sonicator (Sonic Dismembrator Model 100, Fischer Scientific) for approximately 1 min prior to the aggregation assay.

Aggregation assays

Aggregation assays for PAP248-286 were performed in 10 mM phosphate buffer, pH 7.3 containing 150 mM NaCl. A total volume of 40 µL was placed in each well of a 96 well half area plate with a clear bottom, with 2 mg/mL PAP248-286, 25 µM Thioflavin T, and either 0, 1, 2.5, or 5 mol% preformed fibers of curli A or B. Aggregation assays for IAPP and Aβ1−40 were performed similarly except different concentrations of peptide (2.5 µM of IAPP and 5 µM Aβ1−40) and CsgA and CsgB were used and the temperature for IAPP aggregation was set to 25 °C. Amyloid fibers of Aβ1−40 and IAPP were prepared by aggregation for 2 days at 37 °C as detailed below and were sonicated for 1 min before loading onto the plate for the seeding experiments.

A single 1 mm glass bead was placed in each well to increase the aggregation rate and promote sample reproducibility (Giehm & Otzen, 2010). The loaded plate was sealed, and placed on a BioTek Synergy 2 plate reader set at 37 °C (PAP248-286 and Aβ1−40) or 25 °C (IAPP) with a constant linear shaking speed of 17 Hz. Absorbance was monitored at 350 nm and THT fluorescence measurements were taken with an excitation filter at 440 nm with a bandwidth of 30 nm and an emission filter at 485 nm with a bandwidth of 20 nm. Data points were collected every 10 min, with continual shaking occurring between data points. All experiments were performed with samples in triplicate. The kinetic curves were fitted to a sigmoidal growth model that has empirically been found to reproduce most of the features of amyloid aggregation: (1) I=Imax−I01+e(t−t1/2)/k

where I0 and Imax are the initial and maximum fluorescence or absorbance values, t1/2 is the time required to reach half intensity, and the elongation time te is an apparent first order time constant for the addition of peptide to existing fibers (Naiki et al., 1991). The lag-time t0 before detectable aggregation occurs is described by t0 = t1/2−2te.

Transmission electron microscopy

Samples of 6 µL of PAP248-286 solutions after the aggregation experiment were applied to 200 mesh carbon coated copper electron microscopy grids and allowed to stand for 2 min. The grids were then washed with water 20 times to remove salts, after which 6 µL of a 2% uranyl acetate solution was added and allowed to set for 1 min before being removed. Fiber images were taken on a Phillips X-100 Transmission Electron Microscope operating at 60 kV and 10,500 × magnification.

Cells for infectivity assays

A CEM-GFP cell line that expresses a green fluorescent protein (GFP) driven by the HIV-1 LTR promoter was obtained through the AIDS Research and Reference Reagent Program, Division of AIDS, NIAID, NIH from Dr. Jacques Corbeil and maintained in RPMI1640 (Invitrogen) medium supplemented with 10% heat-inactivated fetal bovine serum (HyClone) (RPMI-10) and containing 500 µg/ml geneticin (Invitrogen) (Gervaix et al., 1997).

Infectivity assays

Infectivity assays were performed as previously described (Ono, Monde & Chukkapalli, 2011). A viral stock solution was prepared by transfection of HeLa cells with a HIV-1 molecular clone pNL4-3. Supernatants of transfected cells were collected 2 days post-transfection, and viruses in the supernatant solution were pelleted by ultracentrifugation and resuspended in RPMI-10 medium. These virus stocks (20,000 cpm RT activity) were combined with stock solutions of SEVI amyloid fibers (50 µg/ml from a stock solution of 440 µM PAP248-286 incubated under aseptic conditions for 7 days at 37 °C under vigorous agitation (1300 rpm)) and used to inoculate 2  × 105 of CEM-GFP in 100 µl RPMI-10 for 2 h at 37 °C. Cells were then washed and incubated in 1 ml RPMI-10 at 37 °C. Inoculations were performed in triplicate. To block the second round of infection, the CD4 blocking antibody Leu3a (0.25 µg/ml) (BD Biosciences) and the reverse transcriptase inhibitor AZT (1 µM) (Sigma) were added to the medium 12 h post-infection. Two days post-infection, cells were fixed in 4% paraformaldehyde in PBS (phosphate buffer saline) and analyzed using a FACSCanto flow cytometer and the FlowJo software ver. 8.7.1. Efficiencies of infection were determined directly from the percentage of GFP positive cells after subtraction of the baseline activity obtained in the absence of HIV-1NL4−3. Comparisons between samples were made using a one-tailed unpaired Student t-test.

Results

Seeding with curli fibers greatly increases the rate of SEVI formation from PAP248-286

To test the in vitro activity of curli on the kinetics of SEVI amyloid formation from PAP248-286, we measured the kinetics of amyloid formation and aggregation of 440 µM PAP248-286 solutions in the presence of curli nuclei (Giehm & Otzen, 2010). The curli amyloid fiber is actually a composite of several proteins, (Barnhart & Chapman, 2006) with CsgA serving as the main structural scaffold and CsgB nucleating amyloid formation from CsgA (sequences given in Fig. S1) (Wang, Hammer & Chapman, 2008).

PAP248-286 aggregated slowly in the absence of preformed nuclei of any type under the conditions tested (Olsen et al., 2012), as shown by both turbidity measurements (a nonspecific indicator of general aggregation, Figs. 1C and 1D) and ThT fluorescence (a measurement specific for amyloid, Figs. 1A and 1B). The lag time of formation (∼60 h) is considerably longer than that previously described (∼18 h), (Munch et al., 2007; Ye et al., 2009) at an identical concentration, most likely due to a difference in shaking speed or ionic strength (Olsen et al., 2012).

Figure 1 Kinetics of SEVI amyloid fiber formation in the presence of preformed fibers of CsgA and CsgB.

Top: Turbidity measurements at 350 nm (A) PAP248-286 + CsgA, (B) PAP248-286 + CsgB. Bottom: ThT fluorescence measurements of (C) PAP248-286 + CsgA, (D) PAP248-286 + Csg B. Curves are averages for 3 measurements.

Turbidity increased before ThT fluorescence for samples without CsgA or CsgB (Fig. 1). In addition, while the changes in turbidity could be closely approximated by a sigmoidal curve for all samples, analysis of the residuals from the sigmoidal fit to the ThT fluorescence shows two additional features not present in the turbidity curves. First, a second early component with a short lag time (about 18 h, similar to previous observations (Munch et al., 2007; Ye et al., 2009)) but low ThT fluorescence (about 1/8 of the final value) can be detected in the ThT measurements. Second, ThT fluorescence immediately decreases after the addition of high concentrations of CsgA. These findings suggest amyloid formation by PAP248-286 may be a multistep process in which either the formation of non-amyloid prefibrillar aggregates occurs before amyloid formation for these samples (Suzuki et al., 2012; Horvath et al., 2012) or bundling of amyloid-like protofibrillar filaments is necessary for ThT binding to SEVI, as has been observed for other amyloidogenic proteins (Cabaleiro-Lago et al., 2010).

The addition of preformed CsgA seed had a dramatic impact on the kinetics of SEVI amyloid formation. One mol% CsgA (relative to PAP248-286) was sufficient to cause a six-fold decrease in the elongation time constant, which is reflective of the time for the addition of PAP248-286 to existing fibers (Fig. 2B). CsgA seeds had only a minor effect on the lag time (Figs. 2A and 2C). This is an unexpected result, as the addition of seeds usually results in the reduction or elimination of the lag-time with little corresponding change in the rate of addition to pre-existing fibers (Yagi et al., 2005). CsgB has qualitatively similar effects as CsgA on the kinetics of SEVI amyloid formation, however the magnitude of the effect is relatively less in comparison to CsgA (Figs. 1 and 2).

Figure 2 Elongation of SEVI amyloid fibers is significantly enhanced by preformed fibers of CsgA and CsgB lag time is less affected.

Impact of preformed curli A and B fibers on the lag time (A and C) and elongation time (B and D) of SEVI formation. (A and B): Kinetic constants as determined by ThT fluorescence. (C and D): Kinetic constants as determined by turbidity measurements. Error bars represent S.E.M.

The difference in elongation rates for Csg initiated PAP248-286 fiber formation is reflected in the morphology of individual fibers. Fibers samples initiated by curli and those grown in their absence are morphologically similar, except for a large difference in the aspect ratios (Figs. 3A–3C and Fig. S2). Standard SEVI fibers have an aspect ratio of approximately 5.8, while those grown with CsgA and CsgB are more heterogeneous and much longer, with aspect ratios of approximately 14.7 and 10.8 respectively (Fig. 3F and Fig. S2). This finding is in agreement with the greatly enhanced fiber elongation rate found in the presence of either of the curli fibers. The very short and broken fibers of all PAP248-286 samples are different than typical amyloid fibers, such as the CsgA and CsgB fibers formed under quiescent conditions (Figs. 3D and 3E), most likely because the high degree of agitation required for SEVI fiber formation fragments nascent amyloid fibers.

Figure 3 Curli nucleation produces longer SEVI fibers.

Top: TEM images of SEVI fibers formed in the absence of curli (A) and in the presence of 5 mol% CsgA (B) and CsgB (C) fibers. Bottom: TEM images of CsgA (D) and CsgB (E) fibers. (F) Aspect ratios of individual fibers grown with and without curli nucleation (n = 47, 23 and 37 for SEVI, SEVI + CsgA, and SEVI + CsgB respectively). Fibers were grown at a concentration of 440 µM PAP248-286 at 37 °C under 1400 rpm orbital shaking for 7 days. P values were determined using a two-tailed unpaired Student t-test against the control sample.

SEVI amyloid fibers obtained from curli nucleation retain the ability to enhance HIV infection

We next confirmed that curli nucleated SEVI fiber samples retain a similar ability to promote HIV infection as SEVI fibers generated de novo. The activity of the SEVI fibrils incubated with Csg was tested using an infectivity assay in which a reporter T cell line that expresses GFP upon HIV infection was used (Gervaix et al., 1997). In the absence of SEVI, flow cytometry showed a low percentage of GFP-positive cells, in agreement with the low infectivity of HIV under the conditions employed. The addition of SEVI fibrils generated de novo caused an approximately 8-fold increase in the infectivity (Fig. 4), roughly matching results of other studies of SEVI under conditions of high viral load (Roan et al., 2009; Roan et al., 2010; Hauber et al., 2009). The much larger degree of enhancement shown in some studies ( > 100,000 times) is only apparent at very high viral dilution (Munch et al., 2007). Neither the PAP248-286 monomer nor equivalent amounts of CsgA or CsgB alone had an effect on HIV infection efficiency (Fig. 4). On the other hand, CsgA and CsgB nucleated fibers enhanced HIV infection at least to the same extent as de novo generated SEVI fibers (13 and 16 times respectively, Fig. 4). From these results, it can be seen that the SEVI fiber samples, regardless of how they were nucleated, show a comparable ability to enhance the rate of HIV infection.

Figure 4 Curli nucleated SEVI fibers enhance HIV infectivity to a similar degree as SEVI generated de novo.

CEM-GFP cells were infected with HIV- 1NL4−3 (20,000 cpm RT activity) supplemented either with 50 µg/ml of the proteins indicated or PBS (phosphate buffered saline). SEVI samples were fibrillized for 7 days prior to infection. Measurements were performed in triplicate, error bars indicate S.E.M. P values were determined using a two-tailed unpaired Student t-test against the control sample.

Cross-seeding with curli at low concentrations affects the amyloidogenesis of other proteins besides PAP248-286

The ability of curli to accelerate SEVI formation in the absence of any obvious sequence similarity suggests curli may accelerate amyloid formation by other proteins as well. To test this possibility, we performed analogous seeding experiments on the amyloidogenic peptides IAPP and Aβ1−40. Addition of preformed curli nuclei had a complex effect on the aggregation of both these peptides (Fig. 5 and Fig. S3). The fibrillization rate of IAPP was strongly decreased by low concentrations of both CsgA and Csg B (1% of the IAPP concentration or 25 nM) (Fig. 5). Addition of CsgA, but not CsgB, to IAPP also lowered the total ThT fluorescence, suggesting either fewer or shorter fibers are produced in the presence of CsgA. The effect of CsgA and CsgB on the fibrillization rate of Aβ1−40 was more modest, and showed a more complex dependence on the concentration with the fibrillization rate slightly decreasing at lower concentrations and slightly increasing at higher concentrations of both CsgA and CsgB (Fig. 5). The lag time of both peptides decreased by approximately 50% after addition of 10 mol% of the oppositely charged Csg protein (CsgA for IAPP and CsgB for Aβ1−40), smaller amounts had little effect. Addition of 1% of the similarly charged Csg protein increased the lag time of both peptides, surprisingly, larger amounts had little effect on the lag time. By comparison, addition of equivalent amounts of preformed Aβ1−40 fibril seeds led to a linear decrease in induction time but had little effect on the fibrillization rate (Fig. S4). Although a complete characterization of curli with IAPP and Aβ1−40 lies outside the scope of this study, it can be seen from these experiments that both the effect of curli nucleation on amyloid fibrillogenesis is not limited to PAP248-286 and that curli can serve as both an inhibitor and enhancer of fibrillization.

Figure 5 Curli’s ability to influence amyloid formation is not limited to PAP248-286.

Impact of preformed csgA and csgB fibers on the lag time (A and C) and elongation time (B and D) of amyloid formation from 2.5 µM IAPP (top) and 5 µM Aβ1−40 (bottom) as molar fractions of the IAPP and Aβ1−40 concentrations.

Discussion

In this study, we characterized the kinetics of PAP248-286 cross-seeded by the curli proteins CsgA and CsgB in comparison to the analogous cross-seeding interactions with the more amyloidogenic proteins hIAPP and Aβ1−40. The purpose of this experiment is two-fold. First, PAP248-286 is only biologically active in the SEVI amyloid fiber form. The production of these fibers is ultimately controlled by the rate of amyloidogenesis, as PAP248-286 is subject to inactivating proteolysis in its monomeric but not in its amyloid form (Martellini et al., 2011). Since SEVI fibers have been detected in semen under conditions that would apparently not easily allow aggregation in vitro of SEVI alone, (Munch et al., 2007; Kim et al., 2010) extrinsic factors are a likely source to look for influences on PAP248-286 aggregation. Second, amyloid cross-seeding as a general phenomenon is not well understood, as apparently contradictory results regarding the efficiency and specificity of cross-seeding have been obtained (Yagi et al., 2005; Han, Weinreb & Lansbury, 1995; Westermark, Lundmark & Westermark, 2009; Morales et al., 2010; O’Nuallain et al., 2004; MacPhee & Dobson, 2000; Jarrett & Lansbury, 1992; Hinz, Gierasch & Ignatova, 2008; Rajan et al., 2001). In performing the cross-seeding of PAP248-286, we found that while the lag-time for amyloid formation is moderately affected by curli seeding, the elongation rate is greatly increased. This finding is novel for a cross-seeding reaction and is discussed in more depth below.

An understanding of this result requires some knowledge of the basic mechanism of the cross-seeding reaction. In epitaxial heteronucleation, growth occurs by specific structural matching of the seeding nucleus with the protein being seeded (Fig. 6) (Apostol et al., 2011; Wasmer et al., 2010). A greater understanding of this process can be made by considering the structural constraints for amyloid formation. The cross β-sheet structure common to all amyloids is built by the intermolecular association of β sheets that are stabilized by hydrogen bonds between amide backbone atoms of adjacent sheets. Since the amide backbone, in contrast to the side-chain residues, is similar in all proteins, any unfolded or partially folded protein should theoretically be able to associate with preformed seeds of another to formed mixed fibers. However, amyloid fibers apparently derive much of their energy from the formation of a “steric zipper”, in which the sidechains from adjacent sheets from an interlocking dry surface (Apostol et al., 2011; Sawaya et al., 2007; Eisenberg et al., 2011; Colletier et al., 2011). It can be seen from this requirement that epitaxial heteronucleation is unlikely to occur if the seeding nucleus is structurally different then the protein being seeded, (Apostol et al., 2011; Wasmer et al., 2010; Wiltzius et al., 2009) consistent with the observation that cross-seeding between amyloid proteins is most efficient when the two proteins have homologous sequences (Krebs et al., 2004). For most amyloidogenic proteins this requirement is quite strict and even single point mutations can eliminate the ability of one amyloidogenic protein to cross-seed another. For others, the best studied being α-synuclein which seeds a variety of non-homologous proteins, this requirement is relaxed, possibly because the disorder present in the α-synuclein’s fiber structure can accommodate multiple fiber polymorphs and different fiber interfaces (Yagi et al., 2005).

Figure 6 Cartoon models of possible cross-seeding reactions.

Top: Epitaxial Heteronucleation (A) Binding of the PAP248-286 monomer (red) to the curli seed (blue) induces formation of the β-sheet conformation of PAP248-286 (B) Fiber growth proceeds epitaxially from the seeding nucleus Bottom: Possible mechanism for non-epitaxial heteronucleation (C) A nucleus for the SEVI fiber forms independently of the curli fiber (D) Growth of the SEVI fiber initially proceeds slowly due to unfavorable interactions between subunits of the fiber (E) Lateral attachment of the nascent SEVI fiber to curli reduces repulsion between fiber subunits thereby enhancing the rate of fibrillogenesis. The curli seed may be incorporated into the final SEVI fiber or may detach to catalyze further fiber extension events.

In the simplest model of epitaxially nucleated amyloidogenesis, the elongation time decreases linearly with the concentration of the seed as each new seed provides a new point of fibril growth (Padrick & Miranker, 2002; Cohen et al., 2011). However, the elongation time is less sensitive to the seed concentration in more complicated models explicitly considering secondary nucleation by fragmentation of existing fibers, as the new fibers created by fragmentation provide additional points for fibril growth that compete with the nuclei from the original seeds (Cohen et al., 2011; Knowles et al., 2009). The maximal elongation rate is in fact relatively insensitive to the seeding concentration when the number of nuclei created by fibril fragmentation is much greater than the number of nuclei available from seeding (Cohen et al., 2011; Knowles et al., 2009). Such behavior is typically observed experimentally (Yagi et al., 2005; Han, Weinreb & Lansbury, 1995; Padrick & Miranker, 2002; Xue, Homans & Radford, 2008) and is seen here with the Aβ1−40 protein (see Fig. S4). Regardless of the exact quantitative expression, current models predict that the lag-time for epitaxial heteronucleation should be more strongly affected than the elongation time.

However, epitaxial heteronucleation is not the only possible mechanism by which cross-seeding can occur. In contrast to epitaxial heteronucleation, non-specific heteronucleation can affect both the lag-time and the elongation rate by affecting the stability of different species along the aggregation pathway or by lowering the surface tension associated with forming clusters of the protein. In these mechanisms, the seed is not necessarily incorporated into the new amyloid fiber. Non-specific heteronucleation has been proposed for surfaces and non-protein ligands, (Pronchik et al., 2010; Auer, Trovato & Vendruscolo, 2009; Yin, Chen & Liu, 2009) but not to our knowledge for cross-seeding reactions with other amyloidogenic proteins. Experiments with Lys to Ala PAP248-286 mutants show that charge repulsion between monomer units destabilizes the amyloid fiber (Roan et al., 2009). Lateral association of the curli fiber with the nascent PAP248-286 fiber can reduce this repulsion between monomer subunits, increasing the elongation rate but not affecting the lag-time. In this case, it is expected that the electrostatic differences between CsgA and CsgB (−6 overall charge for CsgA and + 5 for CsgB, PAP248-286 carries an overall charge of + 6) are primarily responsible for the difference in cross-seeding efficiencies, although differences atomic-level differences in the packing between CsgA and CsgB and PAP248-286 may also play a role. It is important to note that a stable interaction between the PAP248-286 amyloid fiber and CsgA or CsgB may not be required, as theoretical studies show that the stability of amyloid fibers increases with the length of the fiber (Linse & Linse, 2011). A transient interaction of CsgA or CsgB with PAP248-286 may stabilize a short, energetically unfavorable PAP248-286 fiber long enough to promote the formation of a longer and more stable amyloid fiber.

The interaction of curli with IAPP and Aβ1−40 is less clear. Electrostatics plays a role in cross-seeding nucleation, as the oppositely charged curli protein reduces the lag-time of both IAPP and Aβ1−40 but the similarly charged curli protein has less effect. Both CsgA and CsgB decrease the elongation rate of IAPP, most likely by binding to and blocking reactive fibril ends. The effect of CsgA and CsgB on the elongation rate of Aβ1−40 has a complex concentration dependence, likely the result of a mixture of stimulatory and inhibitory effects previously observed in the binding of some ligands to amyloid peptides.

Although the effects of curli on Aβ1−40 were moderate, cross-seeding between other bacterial functional amyloids and Aβ1−40 may have greater clinical significance. For example, the bacteria Borrelia burgdorferi produces a curli-like amyloid protein that colocalizes with Aβ amyloid deposits in Alzheimer’s patients (Miklossy, 2008; Miklossy et al., 2004). Similarly, inoculation with Chlamydia pneumoniae stimulates the production of Aβ1−42 amyloid plaques, (Gerard et al., 2006; Little et al., 2004) although to our knowledge a curli-like amyloid protein has not been found yet for Chlamydia pneumonia. Although a definitive link between bacterial infection and amyloid-associated neurodegenerative diseases has been elusive due to the difficulties in firmly establishing bacterial infection considering the low levels of bacteria typical of chronic long-term infections (Dobson, Wozniak & Itzhaki, 2003) the identification of new amyloidogenic proteins in bacteria and mammals is increasing rapidly (Jordal et al., 2009) opening up the possibility of an increasing role for bacterial and viral infections in poorly understood amyloidogenic diseases.

Supplementary Information

Supplementary Information 1 Supporting Information

Click here for additional data file.

The following reagent was obtained through the AIDS Research and Reference Reagent Program, Division of AIDS, NIAID, NIH: CEM-GFP from Jacques Corbeil.

Additional Information and Declarations

Competing Interests

Author Contributions

One of the authors, Ayyalusamy Ramamoorthy, is an Academic Editor for PeerJ.

Kevin Hartman performed the experiments, analyzed the data.

Jeffrey R. Brender and Ayyalusamy Ramamoorthy conceived and designed the experiments, performed the experiments, analyzed the data, wrote the paper.

Kazuaki Monde performed the experiments.

Akira Ono conceived and designed the experiments, analyzed the data, wrote the paper.

Margery L. Evans and Matthew R. Chapman contributed reagents/materials/analysis tools, made other contributions.

Nataliya Popovych made other contributions.

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
