# Peer review of "Bacterial curli protein promotes the conversion of PAP248-286 into the amyloid SEVI: cross-seeding of dissimilar amyloid sequences"

_PeerJ, doi:10.7717/peerj.5_

## Round 0.1 · original submission · Major Revisions

Dear Author, we much appreciate that this is a contribution to the experimental phase of PeerJ and I am sure that the criticisms of referee 1 need to be filtered through this understanding. I would nevertheless ask you to go through the ms and try to accommodate most of the referee's request when possible.

Reviewer 1 ·

Basic reporting

Several aspects of the author-guidelines have not been respected in the manuscript, particularly the citation guidelines have not been respected in the text. Please check the reference style section in the author-guidelines

Minor comments:
-The year of Capule et al is missing in the text and the reference list
-P3 last paragraph: E coli -> Escherichia coli
-The use of italics for e.g. “in vivo” or "de novo" is not consistent throughout the text
-The abbreviations ‘HIV’ and ‘SEVI’ on line 2 and 3 in the abstract should be spelled out.
Idem for IAPP, BCA...
-P3 line 4: it is desirable to express the plural of ‘an additional cofactor’ (like the first line of the next paragraph) unless the authors have indications that only a single cofactor is appropriate

Experimental design

The materials and methods section lacks some essential data and even incorrect information: some methods are described with insufficient detailed information to be fully reproducible by other investigators.

- The source of Abeta40 is not mentioned
- 8M guanidinium HCl: which is the source of the chemical, and how was the pH controlled to pH 7.2?
- Was the protein concentration measured before or after the fibrillation?
- What do the authors mean with “prior to being loaded onto the plate”? It only becomes clear in the next paragraph if my interpretation is correct on how the experiment was actually performed.
- Fiber preparation of abeta40 should not be explained in the section “CsgA and CsgB preparation” but rather inserted in the next section ”aggregation assays”
- Is the concentration of PAP248-286 3mg/mL in the aggregation assay as indicated in the ‘peptide preparation’ section? Or is it 440 µM (2mg/mL) as mentioned in the legend of Fig 3
- The ThT fluorescence is recorded with emission at 440 nm and excitation at 485nm. This is incorrect and actually impossible. Was the fluorescence measured at the top or bottom of the wells?
- What is the operational voltage of the TEM analysis, and at which magnification were the images made?

Minor comment:
Please spell out “BCA“

Validity of the findings

- I find the absence of a characterization of the seeds a missed opportunity to strengthen this work. Are the “seeds” after sonication merely preformed fibers (see p4)? A simple TEM observation would clarify a lot.
-is the non-seeded curve collected with the addition of buffer to the plate (i.e. exactly the same manipulation as the addition of seeds)
- P6 line 6: can the authors rule out that the pretreatment with acetic acid explains the prolonged lag phase?
- That the turbidity increases prior to the increase in ThT fluorescence is difficult to interpret from figure 1, and seems not to be a valid statement for Fig 1B & D. Also in Fig 2A & C this does not seem to be significant. The authors should specify more on how the non-seeded curve fits Equation 1. It seems that the transition in the cross-seeded ThT curve have an initial slow component followed by a cooperative aggregation behavior, while this is not mentioned or described by the authors.
- Is there an explanation for the initial decrease in the ThT fluorescence (in the first 10-20h) with 2.5% and 5% CsgA in Fig 1C?
- P6 last line of 1st paragraph: is more appropriate for the discussion section than the results section
-The description of the fiber morphology is insufficient and should be extended with more details (dimensions, twists…). Aspects of fiber polymorphism and heterogeneity in the samples should also be described. The quality of the images is of insufficient quality / magnification to allow a careful assessment of the statements/interpretation of the authors.
- P6 4th paragraph: is a statement “Csg-containing SEVI fibrils” correct? Did the authors verify that the SEVI Fibrils resulting from the cross-seeding have incorporated Csg proteins in the fibrillar assembly?
-p7 3rd sentence of the discussion: this statement is contradictory to the last sentence of 1st paragraph of page 6

Additional comments

Although I acknowledge the potential in the presented study, I have concerns about the description of the materials and methods and the interpretation of the data. Particularly the materials and methods section should be optimized significantly to allow other investigators to reproduce this work.
More care should have been used while preparing the manuscript for the present submission.

Reviewer 2 ·

Basic reporting

No comments

Experimental design

No comments

Validity of the findings

No comments

Additional comments

This is a very interesting paper which clearly illustrates the complex cross-seeding/elongation effects. I completely agree with the authors: cross-seeding among amyloids with different sequences is common; and like the authors, I expect it is much more common that previously recognized. This aspect has been recently reviewed (e.g. Selective molecular recognition in amyloid growth and transmission and cross-species barriers. J Mol Biol. 2012; 421(2-3):172-84), and emphasized again with respect to tau proteins (Cross-seeding and conformational selection between three- and four-repeat human Tau proteins (J Biol Chem. 2012; 287(18):14950-9; Conformational basis for asymmetric seeding barrier in filaments of three- and four-repeat tau. J Am Chem Soc. 2012 Jun 20;134(24):10271-8). The authors may wish to relate to some of the references noted above which discuss this behavior for a different amyloid species. However, what is of particular interest in this work is that it experimentally goes much beyond these: it shows the complex behavior of sequence species: it may moderately affect the nucleation rate while significantly enhance the growth of fibers from existing nuclei; or it may decrease the lag-time of amyloid formation but strongly inhibit elongation; or have concentration effects, all likely depending on the details sequence matching, and broadly on the environment, including concentration.

Altogether, this work points to an important, yet overlooked aspect of amyloidosis and 'infectivity': cross-seeding is there, and is a reflection of the conformational heterogeneity of amyloids, monomers and oligomers. Further, it can also be expressed by heterogeneous consequences, which can be expected. On the downside, it also points to the difficulties in designing drugs to inhibit amyloid formation.

This work is novel and highly significant, and should be published in PeerJ, and I hope without delay. I expect it would be well-cited, and followed by other such studies, along similar lines. The manuscript can be accepted as is.

---

## Round 0.2 · accepted · Accept

I am satisfied with the changes made.